# Uplink Assisted MIMO Channel Feedback Method Based on Deep Learning

**DOI:** 10.3390/e25081131

**Published:** 2023-07-27

**Authors:** Qingli Liu, Jiaxu Sun, Peiling Wang

**Affiliations:** Communication and Network Laboratory, Dalian University, Dalian 116622, China; sunjiaxu@s.dlu.edu.cn (J.S.); wangpeiling@s.dlu.edu.cn (P.W.)

**Keywords:** massive MIMO, CSI feedback, deep learning, multipath reciprocity

## Abstract

In order to solve the problem wherein too many base station antennas are deployed in a massive multiple-input–multiple-output system, resulting in high overhead for downlink channel state information feedback, this paper proposes an uplink-assisted channel feedback method based on deep learning. The method applies the reciprocity of the uplink and downlink, uses uplink channel state information in the base station to help users give feedback on unknown downlink information, and compresses and restores the channel state information. First, an encoder–decoder structure is established. The encoder reduces the network depth and uses multi-resolution convolution to increase the accuracy of channel state information extraction while reducing the number of computations relating to user equipment. Afterward, the channel state information is compressed to reduce feedback overhead in the channel. At the decoder, with the help of the reciprocity of the uplink and downlink, the feature extraction of the uplink’s magnitudes is carried out, and the downlink channel state information is integrated into a channel state information feature matrix, which is restored to its original size. The simulation results show that compared with CSINet, CRNet, CLNet, and DCRNet, indoor reconstruction precision was improved by an average of 16.4%, and outside reconstruction accuracy was improved by an average of 21.2% under all compressions.

## 1. Introduction

With the development of information technology, fifth-generation (5G) mobile communication has become one of the most popular technologies worldwide. It has fundamentally changed the role of telecommunication technology in society [1]. By the end of 2025, the total number of 5G connections will account for about a quarter of all users [2]. One of the key enabling technologies of 5G is the massive multiple-input–multiple-output (MIMO) system. The massive MIMO system deploys hundreds or thousands of antennas at the base station to increase system capacity and reduce user interference and uses spatial multiplexing to increase network throughput. In order to improve throughput, it is necessary to transmit channel state information (CSI) from the user equipment (UE) to the base station (BS). In a Time Division Duplexing (TDD) massive MIMO system, downlink CSI through channel reciprocity can be obtained and then fed back to the BS. However, in a Frequency Division Duplexing (FDD) massive MIMO system, the user must first estimate the downlink CSI and then feed it back to the BS. In a massive MIMO system, each antenna needs to obtain the corresponding downlink CSI. The number of antennas is considerably high, and the CSI also increases sharply, resulting in additional overhead costs and transmission delays. Therefore, for an FDD massive MIMO system, it is very important to reduce the feedback overhead of CSI.

Initially, a codebook-based method was used to reduce overhead, but the codebook-based method increases the amount of overhead as the number of antennas increases, so it is not suitable for a massive MIMO system. In order to further reduce the CSI feedback overhead and ensure the accuracy of CSI acquisition, compressed sensing (CS) was proposed to reduce the CSI feedback overhead. The CS method reduces the feedback overhead by transforming CSI into a special sparse domain [3,4], but the iterative method for solving equations in the CS method has high complexity and cannot be used to obtain good results. 

With the development of artificial intelligence, methods such as deep learning have been applied in the field of communications, including designing base station beamforming [5], channel estimation [6], symbol detection [7], and CSI feedback [8,9,10,11]. Wen Chaokai et al. [8] proposed CSINet, a method of using deep learning to feed back CSI. This method does not rely on knowledge of the channel distribution. By using deep learning training data to effectively manipulate the channel structure, a higher reconstruction speed can be obtained, proving that compared with CS methods, deep learning is more suitable for CSI feedback tasks. However, CSINet only focuses on the sparsity of the angular delay domain and ignores the spatial correlation, resulting in a sharp drop in resolution at low compression ratios (CR). The authors of [9] designed the digital characteristics of the CSI matrix, saved real and imaginary numbers for training separately, and introduced a convolutional block attention module (CBAM) to suppress noise interference during channel transmission. Good accuracy was obtained. However, because a single-resolution convolution kernel cannot adapt to different compression rates at the same time, feedback accuracy was reduced in the case of high CR. The authors of [10] proposed a multi-resolution neural network channel reconstruction network (CRNet) that can extract information at multiple scales. The CRNet designs two types of convolution kernels to handle high and low CR differently, ensuring reconstruction accuracy. At the same time, the large convolution kernel is optimized to reduce the computational complexity of the network. However, the CRNet’s optimization of the complexity of the convolution kernel also brings about a decrease in accuracy. The convolution kernel is divided into two asymmetric parts to reduce the amount of calculation and increase the accuracy of recovery. The authors of [11] used dilated convolution to enhance the perception range of the convolution field without increasing the size of the convolution to obtain higher recovery accuracy, but because the characteristics of hole convolution and CSI information were not suitable, the CSI information consisted of a set of arrays related to the physical factors of the channel, and the dilated convolution naturally lost part of the information, which reduced reconstruction accuracy. The authors of [12] proposed a neural network structure called CSINet+ for multi-rate compressed sensing that can effectively compress and quantify the CSI matrix. CSINet+ proposes a multi-rate compression method that can solve the problem wherein DL-based methods need to store different training parameters for different compression rates. Although CSINet+ improves the accuracy of feedback, it still suffers from a complex network structure design. The authors of [13] proposed a new full convolutional neural network model called DeepCMC, which sacrificed the use of a full connection layer for compression and used convolutional compression as a whole, effectively reducing the training complexity of the network, and adding a quantized nuclear entropy coding block to reduce the redundancy of code words in the transmission process. A multi-user version was proposed, which can be used for joint training. However, as the number of users increases, the complexity of the model skyrockets. In order to further improve the coding efficiency of CSI feedback, the authors of [14] proposed an efficient DL-based compression framework, dubbed CQNet, to solve CSI compression and codeword quantization kernel recovery under bandwidth constraints. CQNet is directly compatible with other DL-based CSI feedback approaches, and the combination of the two can be used to reduce codeword redundancy. CQNet uses a new non-uniform quantization module that can effectively reduce bits without reducing recovery accuracy. However, CQNet is only a quantitative framework model, so CQNet can yield different performance results for different CSI feedback models and cannot be applied to all methods.

Several points can reduce the overhead. First, as mentioned in [15,16,17], the wireless channel between the base station and the user only has a small angular spread (AS). Due to the small AS and the large dimensionality of the channel, massive MIMO channels exhibit sparsity in the angular domain. Second, due to the angle reciprocity between uplink and downlink [17], the authors of [18] showed that the magnitudes corresponding to the uplink and downlink magnitude exhibit a strong correlation, and the absolute value of the uplink and downlink sparsity also exhibit a positive correlation by separating the real and imaginary parts of CSI. Since it is convenient to obtain uplink CSI in massive MIMO systems, the uplink CSI can be used to assist the downlink CSI in extracting features, thereby reducing the downlink feedback overhead. Compared with [18] using only single-resolution convolution kernels, we propose to use multi-resolution convolution kernels to extract features from CSI matrices with different degrees of sparsity, which can obtain more extracted information and achieve higher feedback accuracy. Finally, the weaker UE needs to reduce the depth of the encoder-side network. Considering the above three points, this paper proposes an uplink-assisted channel feedback method (Complex Uplink Net, CUNet) to improve CSI feedback accuracy. 

Specifically, the contributions of our work are as follows:(1)We propose a complex asymmetric convolution channel feedback network model to replace the traditional convolution model. This model includes two parts: CSI compression and CSI recovery. The multi-resolution convolution kernel effectively improves feedback accuracy;(2)In order to verify that multi-resolution can achieve higher accuracy, we conducted ablation experiments, and the results show that the multi-resolution used by this method is due to the use of single-resolution resolution;(3)To reduce the computational load of the model at the encoder side, we use the magnitude of the uplink at the decoder to assist the downlink CSI recovery from uplink-downlink reciprocity. While deepening the depth of the decoder network, the calculations of the parameters do not significantly increase.

## 2. System Model

This paper establishes a single-cell downlink massive MIMO system model. This system has $N_{t}$($N_{t}\gg 1$) antennas at the BS and $N_{r}(N_{r}=1)$ antenna at the UE. This system uses subcarriers to transmit information in $N_{f}$ Orthogonal Frequency Division Multiplexing (OFDM), and the received signal of the nth subcarrier in the downlink is as follows:(1)$$y_{d}^{\left(n\right)}={h_{d}^{\left(n\right)}}^{H}w_{T}^{\left(n\right)}x_{d}^{\left(n\right)}+n_{d}^{\left(n\right)}$$
where $h_{d}^{\left(n\right)}\in C^{N_{t}\times 1},w_{T}^{\left(n\right)}\in C^{N_{t}\times 1},x_{d}^{\left(n\right)}\in C,n_{d}^{\left(n\right)}\in C$ respectively represent the frequency domain channel vector, precoding vector, transmitted data symbols, and additive white Gaussian noise on the nth subcarrier. ${\left(\cdot \right)}^{H}$ represents the conjugate transpose. The uplink received signal of the nth subcarrier is:(2)$$y_{u}^{\left(n\right)}={w_{R}^{\left(n\right)}}^{H}h_{u}^{\left(n\right)}x_{u}^{\left(n\right)}+{w_{R}^{\left(n\right)}}^{H}n_{u}^{\left(n\right)}$$
where $w_{R}^{\left(n\right)}$ represents the received precoding vector, the element with subscript u represents the vector symbol of the uplink, and the noise is the same as (1). The downlink and uplink CSI matrices in the space–frequency domain are expressed as ${\tilde{H}}_{d}={\left[h_{d}^{1},\cdots ,h_{d}^{\left(N_{f}\right)}\right]}^{H}\in C^{N_{f}\times N_{t}}$, ${\tilde{H}}_{u}={\left[h_{u}^{1},\cdots ,h_{u}^{\left(N_{f}\right)}\right]}^{H}\in C^{N_{f}\times N_{t}}$.

In the FDD system, UE will return to BS through the feedback link $\tilde{H}$, The parameters of feedback in the channel are $N_{t}N_{f}$. In massive MIMO systems, the number of antennas $N_{t}$ is huge, so the number of parameters that need to be fed back is unacceptable for communication systems. To reduce the feedback overhead, the sparsity can be increased by two-dimensional discrete Fourier transformation (DFT) in the angular delay domain.
(3)$$H=F_{d}\tilde{H}F_{a}^{H}$$
where $F_{d}$ and $F_{a}$ are the discrete Fourier transform matrices of dimension $N_{c}\times N_{c}$, $N_{t}\times N_{t}$. Due to the limitation of delay expansion, only the front $N_{c}$ row ($N_{c}\ll {\tilde{N}}_{c}$) of the sparse matrix, H has non-zero elements, and the remaining row elements are almost zero. So keep the previous $N_{c}$ row, and remove the remaining row. Here, we still use $N_{c}\times N_{t}$ to represent the H matrix, so the total number of parameters required for feedback is $2\times N_{c}\times N_{t}$. However, for the downlink feedback, the feedback overhead is still unacceptable. The complex asymmetric neural network designed in this paper will further compress the matrix, input the H matrix into the neural network for compression, and γ generate codeword s according to the compression rate:(4)$$s=f_{en}(H,\Theta_{en})$$
where $f_{en}$ represents the encoding process. After the encoder, the CSI is compressed into an M-dimensional vector codeword s, where M<N, and the compression rate of the data γ=M/N.

When BS receives the codeword s, it inputs s into the decoder for decoding:(5)$$\hat{H}=f_{de}(s,\Theta_{de})$$

$f_{de}$ indicates the decoder. Therefore, the whole feedback process can be expressed as:(6)$$\hat{H}=f_{de}(f_{en}(H,\Theta_{en}),\Theta_{de})$$

The focus of this paper is on the correlation between the reduced feedback matrix and the original matrix $\hat{H}$, which can be identified as the parameter settings of the encoder and decoder in (7):(7)$$\left({\hat{\Theta }}_{en},{\hat{\Theta }}_{de}\right)=\underset{\Theta_{\mathrm{en}}\Theta_{\mathrm{de}}}{\mathrm{arg min}}{\left\|H-f_{de}(f_{en}(H,\Theta_{en}),\Theta_{de})\right\|}_{2}^{2}$$

## 3. Network Structure

### 3.1. Complex-Valued Neural Network Component

The complex-valued network was proposed by the authors of [19], mainly to perform operations equivalent to traditional real-valued 2D convolutions in the complex field, and CSI is also a set of complex values, so compared to traditional real-valued convolutional networks, complex-valued networks are more suitable for CSI feedback. This paper regards complex numbers as a pair of two-dimensional real numbers, expressed as z=A+ib, and divides the input and weight of the network into two parts. During the operation, they are multiplied and then combined according to the combination method in complex number multiplication. The complex value calculation is shown in Figure 1, in which $K_{I},K_{R},M_{I},M_{R}$ respectively represent the imaginary part convolution kernel, the real part convolution kernel, the imaginary part feature map, and the real part feature map.
(8)$$M*K=\left(M_{R}K_{R}-M_{I}K_{I}\right)+i\left(M_{R}K_{I}+M_{I}K_{R}\right)$$

### 3.2. CUNet Structure

As shown in Figure 2, CUNet is divided into two parts: the encoder and the decoder. The performance of the CSI feedback largely depends on the compression part, that is, the encoder. Therefore, it is necessary to improve the CSI performance of the encoder as much as possible, according to the actual situation. Considering the following three situations: (1) In indoor scenes, the CSI matrix is relatively dense and has few non-zero points, which makes it more suitable for convolution with a smaller convolution kernel size to extract fine features. However, in outdoor scenes, CSI becomes scattered and complicated due to the increase of non-zero points; it is better to choose a convolution with a larger convolution kernel size. (2) The mobile device on the UE, that is, the encoder side, has limited computing power, so the depth of the network on the encoder side should be reduced as much as possible. (3) In order to achieve higher accuracy, the convolution block should be improved. CSI is a complex-valued matrix, so it is more suitable to use complex-valued asymmetric convolution. The asymmetric convolution block is shown in Figure 2, taking a convolution block with a convolution kernel size of 3 as an example; the asymmetric module is composed of three convolution kernels with sizes of 1 × 3, 3 × 1, and 3 × 3. The middle strip convolution in the module is to enhance the kernel skeleton of the convolution. Feature information can be extracted more accurately. On the other hand, although two strip convolutions are added to the traditional convolutional network, which increases the number of parameters required in the training phase, in the actual deployment phase, the three convolution sums of the asymmetric module can be equivalent. It can be converted to a standard convolution structure without adding any parameters, so the actual use of asymmetric convolution does not increase any parameters. This means using asymmetric convolutions without additional overhead. At present, its superiority is proved in the CV task [20], and this paper applies it to wireless communication. Therefore, as shown in the Figure 2 encoder design, CSI first extracts features of different sparsities through a layer of multi-resolution layers, because the use of linear rectification function (ReLU) in traditional neural networks will lose negative values, so this paper uses batch one and the leaky linear rectification function (Leaky Relu) after each convolutional layer to solve the problem of gradient disappearance, and then concat through 1 × 1 and 3 × 3 convolution kernels. The CSI is restored to its original size, and the CSI is then compressed.

Considering that the CSI of the uplink and downlink is still partially correlated in FDD, the authors of [18] propose that the CSI amplitude of the uplink can be used to assist the recovery of the downlink CSI, so the CSI of the uplink and downlink information are extracted from the network in parallel. The decoder side is set as two parallel paths. Once the decoder receives the compressed CSI matrix, it will also use the uplink CSI to help recover the downlink CSI. The downlink CSI first restores the received codeword v to its original length through the fully connected layer, and then restores it through two RefineNet. The RefineNet is shown in Figure 2 and consists of four convolutional layers. The convolution kernel size is 3 × 3, and the number of channels is 2, 8, 16, and 2. The specific features are as follows: First, the size of the input and output data of the RefineNet module are the same. Compared with the traditional implementation, our goal is to refine and improve the accuracy rather than reduce the dimensionality, so we expand the number of extracted features in the process of residual to improve the accuracy. Second, in RefineNet, we introduce shortcut connections to pass the raw data directly to the convolved data for superposition. Inspired by deep residual networks [21,22], this method avoids the vanishing gradient problem caused by multiple stacked nonlinear transformations. At the same time, the uplink will also extract features through the convolutional layer. Then the uplink CSI and downlink CSI will be reintegrated into two feature maps through the convolutional layer. Finally, the value is limited to the range [0, 1] by the sigmoid function.

## 4. Simulation Analysis

### 4.1. Simulation Parameter Settings

For comparison with CsiNet, the dataset in this paper was generated by the COST2100 [23] model. All parameters follow the default settings in [23]. In indoor and outdoor scenes, BS is placed in the center of a square area with lengths of 20 and 400 m, respectively, and UE is randomly placed in the square area of each sample, and the specific parameters are shown in Table 1.

### 4.2. Simulation Parameter Settings

Normalized mean squared error (NMSE) is used to represent the difference between the original channel data H and the reconstructed channel data $\hat{H}$. The formula is:(9)$$NMSE=E\{\frac{{\left\|H-\hat{H}\right\|}_{2}^{2}}{{\left\|H\right\|}_{2}^{2}}\}$$

#### 4.2.1. Ablation Experiment

In order to compare the effects of multi-resolution asymmetric convolution, single-resolution asymmetric convolution, and original convolution on feedback accuracy, a simulation experiment is designed and analyzed. The experimental results are shown in Table 2 and Figure 3.

As shown in Table 2 and Figure 3, compared with the original convolution without asymmetry, the accuracy of asymmetric convolution becomes higher as the compression ratio increases. The accuracy of multi-resolution asymmetric convolution is comparable to that of single-resolution asymmetric convolution at a low compression rate, because the features that can be extracted at a low compression rate are limited, and a good effect can be obtained by using a single resolution. When the compression rate increases, multi-resolution asymmetric convolution can obtain better feedback accuracy than single-resolution convolution.

In order to compare the effects of complex asymmetric convolution, complex convolution, asymmetric convolution, and original convolution on feedback accuracy, we designed and analyzed the ablation experiment. The experimental results are shown in Table 3.

As shown in Table 3, compared with the original convolution, asymmetric convolution can enhance the CSI features obtained, while complex convolution extracts the digital characteristics of CSI, so the feedback accuracy of asymmetric convolution and complex convolution reaches a higher level, and Com + Asy represents the simultaneous use of complex and asymmetric convolution. The digital properties of convolution can be utilized at the same time, and the CSI features can be extracted more efficiently, so using both at the same time can obtain better feedback accuracy than using them alone.

#### 4.2.2. Performance Analysis

The neural learning parameters of CUNet, NMSE, and Floating Point Operations (FLOPs) in indoor and outdoor environments have improved by an average of 0.4%, which is the same order of magnitude as other methods, as shown in Table 4 when compared to DCRNet, which has the fewest parameters. Compared with other feedback methods, the performance of CUNet has improved by 16.4% in the indoor environment and 21.2% in the outdoor environment. The advantage is more obvious under the outdoor high compression ratio, which has increased by 27.6%. Because at high compression ratios, since more features are lost in the compression process, the auxiliary recovery of the uplink is more important. The FLOPs of CUNet are not the smallest. Compared with traditional neural network models, CUNet is still a simple model, such as ResNet50, with FLOPs of 3.9 G, so the additional computational overhead is acceptable, and CUNet’s FLOPs are still in the normal range. In summary, CUNet improves the accuracy of CSI feedback without significantly increasing the number of model parameters. Therefore, CUNet is better suited for feedback tasks.

In addition, as shown in Figure 4, as the number of training increases, the loss of CUNet decreases more stably than the fluctuation of other methods’ loss values, and when CsiNet and DCR Net have reached their peaks, the loss of CUNet still decays slowly until finally converged. It shows that CUNet is more suitable for the CSI feedback task.

In order to verify that the method is also applicable to other channel environments, this paper simulates the massive MIMO Rayleigh flat fading channel model through MATLAB R2021a, and the specific parameter settings are shown in Table 5. The simulation data is divided into training sets, check sets, and test sets, with sample numbers of 100,000, 20,000, and 10,000, respectively.

The comparison between CUNet and CSINet in Rayleigh fading channels is shown in Table 6. It can be seen that the NMSE values of the two methods are basically similar at low CR, because the lower CR is, the fewer data are output in the compression stage, and the original data cannot be accurately reconstructed with fewer data, so the NMSE performance is basically similar. However, in the case of high CR, the proposed CUNet is still superior to CSINet, so CUNet is also suitable for Rayleigh fading channels.

## 5. Conclusions

An uplink-assisted complex neural network method is proposed in this paper. This method changes the original convolutional network to a complex asymmetric convolutional layer that is more suitable for complex-valued CSI matrices and adds an uplink-assisted CSI recovery. Performance experiments show that compared with CSINet, CRNet, CLNet, and DCRNet, the indoor average reconstruction accuracy increased by 16.4%, and the outdoor average reconstruction accuracy increased by 21.2%. Therefore, the CUNet proposed in this paper is more suitable for CSI feedback tasks.

## Figures and Tables

**Figure 1 entropy-25-01131-f001:**
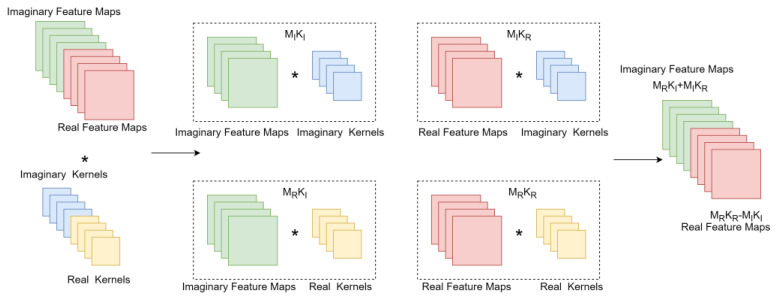
Complex convolutional network component. * is a convolution operator.

**Figure 2 entropy-25-01131-f002:**
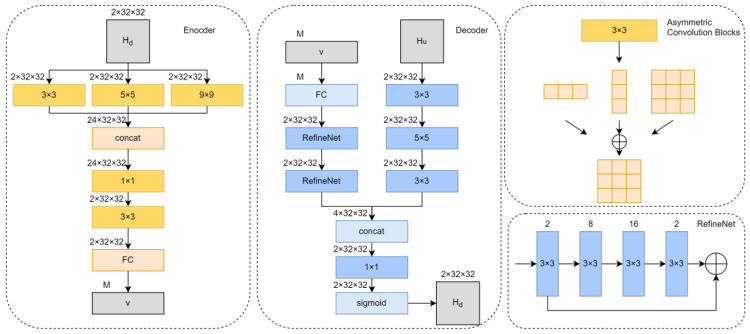
CUNet Structure.

**Figure 3 entropy-25-01131-f003:**
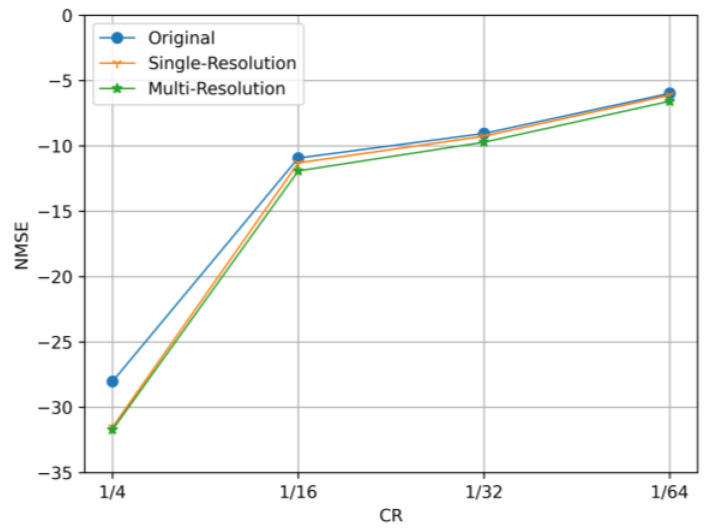
Comparison of NMSE (dB) ablation experiments in the indoor scene.

**Figure 4 entropy-25-01131-f004:**
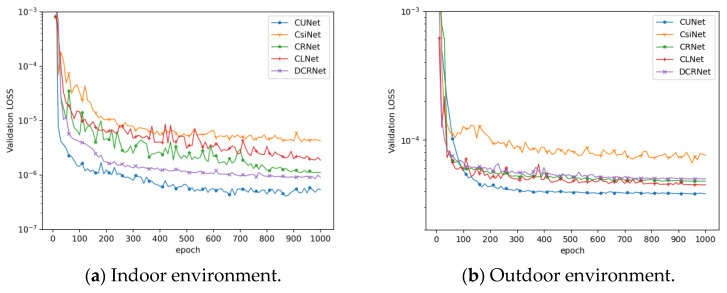
Convergence speed of indoor and outdoor scene network models when the compression rate is 1/4.

**Table 1 entropy-25-01131-t001:** Simulation parameters.

Parameters	Indoor		Outdoor	
	Uplink	Downlink	Uplink	Downlink
Bandwidth	20 MHz		20 MHz	
Subcarrier	1024		1024	
Frequency	5.1 GHz	5.3 GHz	260 MHz	300 MHz
Optimizer	Adam		Adam	
Batch size	200		200	
Training sets	100,000		100,000	
Validation sets	30,000		30,000	
Testing sets	20,000		20,000	

**Table 2 entropy-25-01131-t002:** Comparison of NMSE (dB) ablation experiments in the indoor scene.

CR	Original	Single-Resolution	Multi-Resolution
1/4	−28.02	−31.53	−31.71
1/16	−10.93	−11.30	−11.91
1/32	−9.05	−9.26	−9.72
1/64	−5.98	−6.12	−6.58

**Table 3 entropy-25-01131-t003:** Comparison of NMSE (dB) ablation experiments in the indoor scene.

CR	Original	Asymmetric	Complex	Com + Asy
1/4	−28.02	−30.27	−29.01	−31.71
1/16	−10.93	−11.25	−10.98	−11.91
1/32	−9.05	−9.23	−9.11	−9.72
1/64	−5.98	−6.06	−6.03	−6.58

**Table 4 entropy-25-01131-t004:** Simulation parameters, NMSE(DB), and FLOPs, where CR is the compression ratio.

CR	Method	Indoor	Outdoor	FLOPs	Parameters
1/4	CsiNet	−17.36	−8.75	5.41 M	2103 K
	CRNet	−24.10	−12.57	24.57 M	2102 K
	CLNet	−29.16	−12.88	4.05 M	2101 K
	DCRNet	−28.04	−12.70	4.01 M	2102 K
	CUNet	−31.71	−13.83	4.62 M	2102 K
1/16	CsiNet	−8.65	−4.51	3.84 M	530 K
	CRNet	−10.52	−5.36	23.00 M	/
	CLNet	−11.15	−5.73	2.48 M	/
	DCRNet	−11.74	−5.36	2.44 M	528 K
	CUNet	−11.91	−6.69	3.16 M	530 K
1/32	CsiNet	−6.24	−2.81	3.58 M	268 K
	CRNet	−8.90	−3.16	10.16 M	/
	CLNet	−8.95	−3.49	2.22 M	/
	DCRNet	−9.05	−3.47	2.18 M	266 K
	CUNet	−9.72	−3.61	2.83 M	267 K
1/64	CsiNet	−5.84	−1.93	3.45 M	137 K
	CRNet	−6.23	−2.19	22.61 M	136 K
	CLNet	−6.43	−2.86	2.09 M	135 K
	DCRNet	/	/	/	/
	CUNet	−6.58	−2.97	2.51 M	136 K

**Table 5 entropy-25-01131-t005:** Simulation parameters setting.

Parameters	Setting	
Transmitting antenna	32	64
User	1	
Receiving antenna	8	
Modulation	QPSK	
Antenna spacing	0.5λ	

**Table 6 entropy-25-01131-t006:** NMSE(DB), where CR is the compression ratio.

CR	Method	$N_{tr}=32$	$N_{tr}=64$
1/4	CsiNet	−16.13	−17.53
	CUNet	−19.77	−20.01
1/16	CsiNet	−10.06	−10.81
	CUNet	−11.08	−11.36
1/32	CsiNet	−6.11	−8.04
	CUNet	−6.30	−8.33
1/64	CsiNet	−4.38	−6.17
	CUNet	−4.42	−6.31

## Data Availability

The processed data required to reproduce these findings cannot be shared as the data also form part of an ongoing study.

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
