# Peer review of "Uplink Assisted MIMO Channel Feedback Method Based on Deep Learning"

_entropy, 2023, doi:10.3390/e25081131_

Round 1

Reviewer 1 Report

This paper proposed a deep learning based approach to compress the MIMO CSI, taking advantage of the uplink CSI to reduce the corresponding downlink CSI feedback overhead. The proposed ideas are interesting and the contribution looks sound. The topic has gained much interest recently and can attain sufficient visibility if published. However, the following points should be addressed before publication:

1-Please provide all contributions of the paper in a separate paragraph in the introduction. Specifically, provide a comparison with [15] in design or performance.

2-Please consider better use of space. Figures 2 and 3 are less informative but take too much space. Please merge them into figures 1 and 4 as appropriate.

3-Please provide a complexity analysis in terms of the number of trainable NN parameters.

4-What are the challenges of implementing this design in practice? Are there any downsides to this design? Can you comment on how to deal with the changing wireless environments and channel statistics?

5-DL-based CSI feedback has been receiving much interest recently. I think the current literature review is lacking. Please provide a more comprehensive review, specifically addressing more ComSoc transaction papers like the following seminal works:

[1] J. Guo, C.-K. Wen, S. Jin, and G. Y. Li, “Convolutional neural networkbased multiple-rate compressive sensing for massive MIMO CSI feedback: Design, simulation, and analysis,” IEEE Trans. Wireless Commun., vol. 19, no. 4, pp. 2827–2840, Apr. 2020.

[2] M. B. Mashhadi, Q. Yang and D. Gündüz, "Distributed Deep Convolutional Compression for Massive MIMO CSI Feedback," in IEEE Transactions on Wireless Communications, vol. 20, no. 4, pp. 2621-2633, April 2021.

[3] Z. Liu, L. Zhang and Z. Ding, "An Efficient Deep Learning Framework for Low Rate Massive MIMO CSI Reporting," in IEEE Transactions on Communications, vol. 68, no. 8, pp. 4761-4772, Aug. 2020.

6-The quality of English language should be significantly improved. There are also style and presentation issues. For example, we do not need an indention after equations (1)-(7). We are not starting a new paragraph. 

As the above points take time to address, I suggest a major revision.

The quality of English language should be significantly improved. There are also style and presentation issues. For example, we do not need an indention after equations (1)-(7). We are not starting a new paragraph. 

Please have the paper proofread for English, style, and presentation issues by an expert.

Author Response

Please see the attachment。

Reviewer 2 Report

This paper addressed an uplink-assited MIMO channel feedback method based on DL. This paper is well written and the performance of the proposed technique is excellent. However, since various channel environments are not considered, it is questionable whether the technique will have a performance gain in the actual channel environment. Does the author think there will also be performance gains in Rayleigh fading? It would be nice to add some channel environments and performance evaluation.

Correct some typos.

Author Response

Please see the attachment。

Round 2

Reviewer 1 Report

The authors have well addressed all the reviewer comments.

Reviewer 2 Report

The paper was well revised by reflecting the review comments. I recommend publishing this paper.